# An Analysis of the Factors Associated with the Seasonal Variability of Physical Activity in Natural Environments in a Sample of Lithuanian Adults

**DOI:** 10.3390/bs15060773

**Published:** 2025-06-03

**Authors:** Rasa Jankauskiene, Migle Baceviciene

**Affiliations:** Health Research and Innovation Science Center, Faculty of Health Sciences, Klaipeda University, 92294 Klaipeda, Lithuania; rasa.jankauskiene@ku.lt

**Keywords:** outdoor physical activity, green exercise, seasonal variability, nature exposure, equity

## Abstract

Engagement in physical activity (PA) in natural environments is known to promote physical and psychological well-being, yet little is known about how such activity fluctuates across seasons and how it relates to individual characteristics and quality of life (QoL). This study aimed to assess the seasonal variability of PA in nature and examine its associations with sociodemographic factors, nature-related experiences, and overall QoL in a sample of Lithuanian adults. A total of 924 participants (680 women and 244 men) with a mean age of 40.0 ± 12.4 years completed an online survey. The study measures included sociodemographic characteristics, nature proximity, exposure, connectedness, perceived restoration in nature and QoL measured by the WHOQOL. Based on self-reported seasonal behaviour, participants were categorized into three groups: irregular or no PA in natural environments, seasonal variability, and regular year-round PA in natural environments. In the present study, we observed the lowest rates of PA in natural environments in winter and the highest in summer. Comparative analyses revealed that individuals with regular PA in natural environments reported significantly greater financial security, nature exposure and connectedness, proximity to green spaces, and overall QoL compared to less regular PA in nature groups. A multiple regression analysis identified financial security, nature proximity, nature exposure, connectedness, and perceived restoration in nature as significant and positive predictors of QoL. These findings point to the problem of inequity and suggest that inequitable access to PA in natural environments manifests as a higher seasonality of PA in nature. The practical implications of the study highlight the importance of addressing equity and promoting consistent PA in nature throughout seasons. During the dark, wet, and cold periods, it is important to implement targeted interventions that improve access to natural environments for individuals with lower financial security. This could help reduce inequity in physical activity. Promoting access to green spaces and fostering nature connectedness may be particularly valuable strategies in public health interventions aimed at enhancing QoL across diverse populations.

## 1. Introduction

### 1.1. Physical Activity in Natural Environments and Its Restoration Effect

A considerable body of research has yielded compelling evidence supporting the notion that exposure to natural environments exerts a positive effect on human health and well-being ([9]; [19]; [34]; [39]; [65]). In an objective sense, natural environments refer to the physical characteristics and processes that originate from nonhuman sources and are typically observable by people. This includes the living nature of plants and animals, as well as bodies of water, air quality, weather conditions, and the landscapes shaped by geological activities ([19]). The term “nature” significantly overlaps with the “natural environment”, which is an area with minimal or no visible signs of human presence or interference. The contemporary trend of reduced exposure to natural environments and augmented time spent in indoor settings, utilizing electronic devices, is associated with a diminished sense of connectedness with the natural world ([49]). A recent international study of environmental programmes students in 41 countries found that students in wealthier countries tend to have weaker relationships with nature than those in less affluent countries ([30]). Other findings also suggest that when countries become wealthier, individuals may have less direct connectedness to nature ([10]; [48]), potentially due to urbanization, increased reliance on technology, and lifestyle changes.

In their novel review, Barragan–Jason and colleagues observed experimental studies in various geographical regions and concluded that physical connection with nature improved human cognition, social skills, physical and mental health, and positively affected psychological connection to nature. Correlational studies also identified positive relationships between psychological connection with nature and both mental and physical health ([2]). The psychological connection with nature is defined as the extent to which individuals perceive themselves as being intertwined with the natural world ([60]). In contrast, the physical connection is characterized by the degree of interaction with natural environments ([2]). Finally, evidence suggests that exposure to nature fosters a greater body appreciation, a sense of savouring life, and enhances the sense of living a meaningful life ([57]; [74]).

Physical activity (PA) is one of the forms through which people can establish or maintain a connection with nature and experience the psychological and physical benefits that nature provides ([19]). PA has been identified as a key mediator of the association between nature exposure and mental health ([68]) and between the quality of residential environment and restorative perceptions of the settlement’s environment ([73]). There are three general categories of activities that fall under the umbrella of PA in nature. The first domain, “work”, includes all activities people do at work in natural environments (i.e., gardening). The second domain is “active transport”, which refers to getting to and from work by walking or biking in nature. The third domain is “leisure recreation”, which includes any activities humans do for fun, like sports and other hobbies (i.e., recreational cycling) in natural environments ([19]). A body of research has indicated that PA in natural environments (or green exercise) is associated with enhanced mental health benefits, including relaxation, positive affect, increased energy, enjoyment of nature, lowered stress and anxiety, and general well-being ([5]; [23]; [33]; [42]). These benefits appear to exceed those associated with PA in indoor settings ([33]; [71]). One of the recent explanations why green exercising is more restorative compared to indoor exercise is that nature provides high variability. Specifically, exercising in nature is related to higher attention and holistic cognitive and emotional involvement, meeting various challenging situations towards the environment than in indoor exercising ([1]).

Two major theories have been utilized to explain the positive impact of nature on human health. Stress Recovery Theory (SRT), proposed by Roger Ulrich, is based on the idea that exposure to nature can have positive physiological and psychological effects, helping individuals recover from stress more effectively than in urban environments ([67]). Attention Restoration Theory (ART) suggests that exposure to natural environments can help restore human ability to focus and concentrate ([27]). Specifically, in urban settings, humans use effortful attention for tasks requiring concentration. On the contrary, natural environments contain elements that capture human attention effortlessly, allowing directed attention to rest and recover ([26]; [41]). A systematic review concluded that working memory, cognitive flexibility, and attentional control may be improved after exposure to nature through the restoration of cognitive processes related to directing attention ([56]). For PA in nature, the variability of nature helps to be more mindful and to shift attention from stressful everyday situations that are usual in urban settings towards nature; therefore, exercise in nature might provide more psychological restoration compared indoor exercise ([1]).

### 1.2. Seasonal Variability of PA in Natural Environments

In contrast to indoor exercise, the practice of PA in a natural environment is directly dependent upon meteorological factors and weather conditions such as extreme temperatures, wind, and precipitation level ([16]; [59]). Levels of PA fluctuate with the seasons, and the impact of poor or extreme weather has been identified as a significant barrier to participation in PA among various populations ([63]). Temporal factors such as season and daylight-saving transitions influence the movement behaviours of humans, and in temperate climates, people’s health and quality of life are more at risk during the winter months than during the summer ([21]; [47]). A recent meta-analysis explored 24 h movement behaviours and suggested evidence of seasonal variation in activity patterns, with sedentary behaviour highest in winter (+19 min), light PA higher during spring and in summer (+19 min), and moderate-to-vigorous PA higher during spring and highest in summer (+2 min) when compared to the yearly means ([15]). The study in China indicated that the thermal comfort of green spaces positively affects various aspects of PA, including longer outdoor PA ([40]).

Seasonal variations in PA have important implications for human health ([54]). In most developed societies, in parallel with PA, measurements of fitness indicators, including maximal oxygen intake and muscle strength, commonly increase during spring and summer, while body fat and changes in blood lipids, blood pressure, and blood coagulability increase during the winter ([54]; [61]). Further, all-cause and cardiac mortalities increase in the winter in economically developed countries ([54]). Finally, there is evidence that day length and temperature might have a significant moderating effect on the change in MVPA during a PA intervention. Specifically, the PA intervention was less effective on colder and on shorter days ([70]). Thus, decreasing PA seasonality and determining factors that influence it is an important public health task.

Reducing the seasonality of PA is an issue that previously received scientific attention. For example, the results of a study in Germany showed that participation in a sports club and a commercial sports facility were associated with reduced seasonal variation in PA and a significantly higher probability of participating in PA at a volume above the World Health Organization (WHO) threshold across all seasons ([52]). However, while substituting outdoor activities with indoor activities might be a solution for urbanized regions, this strategy might be complicated for people living in rural or remote places, since the availability of sports facilities might be limited. Further, replacing nature-based PA with indoor exercising might decrease the psychological restoration effect that is related to PA in natural environments. Finally, not all individuals who like to be physically active in nature would like to change their PA (i.e., gardening, working or bicycling in nature) to exercise in sports clubs and or commercial facilities because of possible differences in the motivational goals of these activities. This is especially true for populations of older ages. Therefore, it is important to understand factors that are related to the seasonal variability of outdoor PA. Exploring these factors might inform public health programmes aiming to decrease the seasonality of PA in natural environments and the quality of life of societies.

In most of the research, the seasonal variability of PA or movement behaviours was measured without distinction between organizational settings ([52]); therefore, it is difficult to understand the seasonal variations in PA in natural environments. However, there are some studies that report seasonal variability in different domains of outdoor PA. For instance, a study using a sample of Finnish golf players reported that all participants increased their total PA, demonstrated moderate walking activity (*p* < 0.001), and sat less (*p* < 0.001) in the summer season as compared with the winter season ([28]). Recent research also showed that time spent on low PA and outdoor gardening is decreasing in winter and increasing in spring and summer in older individuals living in mountainous agricultural areas of Japan ([38]). Research in Brazil concluded that more frequent use of public open spaces, including green spaces, was observed during the spring season, and moderate to vigorous PA (MVPA) was more frequent during the winter, at lower temperatures and thermal sensations ([46]). The seasonal variability of cycling was also observed in New Zealand ([62]). Finally, a qualitative study reported that seasonal changes to the winter recreation environment led to a qualitatively different outdoor recreation experience. According to respondents, winter recreation settings can be thematically characterized as paradoxical, novel, wilder, emptier, non-normative, and providing additional outdoor recreation opportunities ([17]). Summing it all up, the impact of seasonal variations and meteorological factors on PA conducted in outdoor spaces remains an understudied area ([50]; [63]).

Finally, previous studies reported that socioeconomic status is an important variable related to seasonal variability of PA in nature. It is well known that nature availability in living areas is associated with greater PA in natural environments ([6]; [45]). PA in natural environments is free of charge and therefore is more available for socially deprived groups compared to organized sports and gyms. Nevertheless, socially deprived groups usually live in places that are distant from natural environments and are perceived as less safe ([51]), more polluted, and noisy ([14]). The results of the previous study suggest that lower socioeconomic status is associated with a greater seasonal variability of exercise ([13]). However, more empirical data are necessary to better understand if the seasonal variability of PA in nature is related to financial security. These data might help to inform policies related to decreasing health inequalities.

### 1.3. The Present Study

Given the evidence that nature exposure has a positive effect on human health and that health effects depend on continuous participation in PA, it is important to identify factors associated with the seasonal variability of PA in natural environments. It is unclear how the seasonal variability of PA in natural environments is associated with various demographic and nature-related variables such as green-space proximity in the living area, nature exposure, nature connectedness, perceived restoration in nature and quality of life. Understanding the interrelationships between these factors might help to better understand how to effectively promote PA in nature and to tailor interventions during colder, wetter, and darker periods.

The aim of the study was to assess the seasonal variability of PA in natural environments and its relationships to demographic, nature-related variables, and quality of life in a sample of Lithuanian adult women and men. We hypothesized that PA in natural environments will be lowest in winter and highest in summer. Second, we expected that the higher seasonality of PA in nature will be related to lower socioeconomic status, green-space proximity, nature exposure, nature connectedness, perceived restoration in nature, and general quality of life.

## 2. Materials and Methods

### 2.1. Study Design and Procedure

This study was approved by the university Social Research Ethics Committee (Protocol No SMTEK-60) and implemented between January and April 2021. The present study is a part of a more extensive international study, “Body Image and Nature Survey (BINS)” ([58]). The study employed a cross-sectional survey design. The survey was conducted using an online questionnaire hosted on the Google Forms platform. Participants were recruited through a non-probabilistic sampling approach, and participation was entirely voluntary and uncompensated. The survey link was distributed through a sponsored Facebook advertisement targeting all major municipalities across the country. Eligibility criteria included being 18 years of age or older and fluency in the Lithuanian language. Before beginning the questionnaire, participants were presented with an overview of the study’s objectives, estimated time required to complete the survey (approximately 25–30 min), and assurances regarding anonymity. Participants provided digital informed consent before proceeding to the survey questions; those who did not consent were thanked and withdrawn from the survey automatically. Furthermore, respondents were free to withdraw from the survey at any time by closing their browser, and no responses would be recorded in such cases. The final sample included 924 Lithuanian adults (both women and men) with complete responses, as all questions in the survey were required to proceed.

### 2.2. Study Participants

A total of 924 Lithuanian adults participated, with an age range between 18 and 79 years (mean age 40.0 ± 12.4 years). The sample consisted predominantly of women (73.6%), and 26.4% were men. Regarding educational attainment, 11.3% had secondary education or less, 7.7% were current full-time students, 41.7% held an undergraduate degree, and 34.8% had postgraduate education; 69.9% resided in urban areas, while the rest resided in rural areas. Marital status distribution showed 18.4% were single, 17.2% were in a committed relationship, and 57.8% were married. Most participants (91.1%) identified themselves as part of the ethnic majority. Concerning financial security, 17.1% perceived themselves as financially less secure compared to others, 64.8% as equally secure, and 18.1% as more secure. Mean body mass index was 24.8 ± 4.6 kg/m^2^ and ranged from 16.4 to 44.9 kg/m^2^, and 3.8% were classified as underweight, 55.7% as normal weight, 28.3% as overweight, and 12.2% as obese.

### 2.3. Measures

Sociodemographic data measures included education assessed via the highest education level completed (secondary, in full-time studies, undergraduate, postgraduate); marital status classified as single, in a committed relationship, married, or other; ethnicity categorized as ethnic majority, ethnic minority, or not sure, with place of residence (urban or rural); and financial security assessed based on self-perceived financial security compared to others (less secure, like most, and more secure).

Body mass index (BMI) was calculated using self-reported weight and height, applying the standard formula: weight, kg/height, m^2^. Participants were then categorized into four groups according to the World Health Organization (WHO) classification ([72]): underweight: BMI < 18.5 kg/m^2^, normal weight: BMI 18.5–24.9 kg/m^2^, overweight: BMI 25.0–29.9 kg/m^2^, and obese: BMI ≥ 30.0 kg/m^2^. This classification was used to describe the sample and explore group differences in BMI across genders.

Nature proximity was assessed using a single-item question asking participants to indicate the distance from their place of residence to nearby natural environments, such as parks or forests. Respondents selected one of the following options: more than 10 km, 5–10 km, 2–4 km, 1–2 km, and nearby (within 0–1 km). Responses were coded so that higher scores reflected greater proximity, indicating a shorter distance from the place of residence to natural environments.

The frequency of PA in nature was assessed by a single question: “How often do you engage in PAs such as exercising, walking, cycling, or performing physically demanding tasks in natural settings (e.g., forest, park)?” Participants responded using a 6-point Likert scale ranging from 1 (never or very rarely) to 6 (every day), with intermediate options indicating varying frequencies (2–3 times a month, once a week, 2–4 times a week, 5–6 times a week). This item was adapted from a previously used measure in a Lithuanian national survey, with modifications to focus specifically on activity in natural environments ([31]). To account for seasonal variation, respondents were asked to rate their frequency of PA in nature separately for spring, summer, autumn, and winter. A mean score across the four seasons was then computed for use in analyses. The scale demonstrated excellent composite reliability, with McDonald’s ω = 0.947 (95% CI: 0.941–0.952). Next, based on the frequency of PA in nature across different seasons, all respondents were classified into three groups. Group 1 included non-exercisers or those engaging in irregular and infrequent PA in nature, defined as selecting response options 1 (never or very rarely) or 2 (2–3 times per month) consistently across all four seasons. Group 3 comprised regular all-year exercisers, who reported engaging in PA in nature at least once a week (response options 3 to 6) in every season. Group 2, labelled the seasonal variability group, consisted of participants whose responses did not meet the criteria for either group 1 or group 3, typically those reporting regular PA in nature during summer but avoiding it in winter. As there were minimal differences between groups 1 and 2, they were combined in the final step of the regression analysis.

Total PA was assessed based on the frequency of moderate-to-vigorous PA lasting at least 30 min ([31]). Participants responded on a scale from 1 (never or very rarely) to 6 (every day), with separate assessments for winter and summer periods. The summer season was defined as the end of spring, summer, and the beginning of autumn, while the winter season was defined as the end of autumn, winter, and the beginning of spring. Additionally, the Godin Leisure-Time Exercise Questionnaire (LTEQ) was used to evaluate PA during the winter and summer seasons ([18]). Participants were asked to report the number of weekly sessions lasting at least 15 min for light, moderate, and vigorous exercise. According to the questionnaire guidelines, each session was weighted by a corresponding intensity coefficient—3 for light, 5 for moderate, and 9 for vigorous activity—and then summed to calculate a total score, reflecting the overall level of leisure-time PA per week in summer and winter seasons.

Nature exposure was assessed using the Nature Exposure Scale (NES) ([24]). The NES has demonstrated unidimensional factor structure and good internal consistency in previous studies examining the psychological benefits of nature contact, and for Lithuanian translation ([35]). The scale consists of four items designed to measure individuals’ subjective exposure to natural environments in their everyday and not everyday lives, and the extent to which natural surroundings are noticed. Participants rate how frequently they engage with nature using a 5-point Likert scale ranging from 1 (not at all) to 5 (very often). Higher average scores indicate greater self-reported exposure to natural environments. In the present study, the scale showed acceptable composite reliability (McDonald’s ω = 0.697, 95% CI = 0.665–0.728).

Connectedness to Nature Scale (CNS) was used to measure connectedness to nature ([36]). CNS is a 14-item self-report instrument designed to assess individuals’ affective, experiential sense of emotional connection with the natural world. Items reflect the extent to which people feel part of nature and identify with it. Responses are rated on a 5-point Likert scale ranging from 1 (strongly disagree) to 5 (strongly agree), with higher average scores indicating a stronger connectedness to nature. The CNS has demonstrated good internal consistency and unidimensional factor structure in our previous study ([35]). In the present study, the scale showed excellent composite reliability (McDonald’s ω = 0.906, 95% CI = 0.897–0.915).

Restorative experiences in nature were assessed using the Restoration Outcome Scale (ROS) ([32]). This scale includes nine items that capture various psychological outcomes associated with recent visits to natural environments. The items are rated on a 7-point Likert scale from 1 (not at all) to 7 (completely). The overall higher score calculated by averaging the response options indicates a greater restorative effect during the last visit to natural environments. The unidimensional factor structure and good psychometric properties were confirmed in our previous study ([35]). For this study, the composite reliability was also high (McDonald’s ω = 0.977, 95% CI = 0.974–0.979).

The World Health Organization Quality of Life Questionnaire–BREF (WHOQOL-BREF) was used to assess participants’ perceived QoL ([55]). The psychometric characteristics of the translated instrument were previously confirmed in the Lithuanian adult population ([12]). The instrument comprises 26 items, including two general items that assess overall QoL and general health, and 24 items that measure four specific domains: physical health, psychological well-being, social relationships, and environment. Responses are rated on a 5-point Likert scale, with higher scores indicating better perceived QoL. According to the recommendations of the WHOQOL-BREF group, the final score of the overall QoL was calculated in a range between 0 and 100, with the higher score indicating better QoL. Domain-specific scores were not included in the analysis. The WHOQOL-BREF has demonstrated good composite reliability in this study (McDonald’s ω = 0.924, 95% CI = 0.916–0.931).

### 2.4. Statistical Analysis

Power analysis using G*Power version 3.1.9.2 indicated that the sample size of 924 was sufficient to detect small-to-medium effects (f^2^ = 0.02–0.15) in multiple regression analyses with up to 15 predictors (α = 0.05, power = 0.80). The overall regression model explained 30% of the variance in overall QoL, corresponding to a large effect size (f^2^ = 0.43) according to Cohen’s criteria ([8]). The sample size also provided adequate power for between-group comparisons using ANOVA.

Statistical analyses were conducted using JASP software version 0.19.3 obtained from the official website (JASP Team: Amsterdam, The Netherlands) and SPSS version 29 (IBM Corp., Armonk, NY, USA). Assumptions of normality of continuous variables were checked and met. The composite reliability of the scales was tested by McDonald’s ω ([20]). Group comparisons by gender and by seasonal patterns of PA in nature were performed using independent samples t-tests for continuous variables and chi-square (χ^2^) tests for categorical variables. Differences in PA between the summer and winter seasons were analyzed using a paired-sample t-test. To explore differences in study variables across the three identified seasonal PA in nature patterns (i.e., irregular/no PA, seasonal variability, regular PA), one-way analysis of variance (ANOVA) was employed. Post hoc comparisons with Bonferroni adjustments were conducted to identify specific group differences, with superscripts indicating statistically significant pairwise differences.

Finally, a multiple linear regression analysis was conducted to examine the predictors of overall QoL. Homoscedasticity was tested using the Breusch–Pagan test in SPSS. This test evaluates whether the variance of the residuals is dependent on the values of the independent variables, thereby indicating the presence or absence of heteroscedasticity ([4]). The test was not statistically significant, indicating that the assumption of homoscedasticity was met. Additionally, a visual inspection of residual plots did not reveal any systematic pattern. The model included sociodemographic variables (e.g., gender, age, financial security, education, marital status, residence), health-related variables (BMI), and nature-related measures (e.g., nature proximity, nature exposure, connectedness, perceived restoration), along with the seasonal PA in nature pattern. Continuous predictors were standardized to compute β coefficients, while categorical variables were entered as dummy-coded predictors with designated reference groups. The level of significance was set at *p* < 0.05.

## 3. Results

Sample characteristics with the comparison in gender groups are presented in Table 1. Women were more likely to have higher levels of education (postgraduate: 40.7% vs. 24.2%, *p* < 0.001). Urban residency was more common among women (71.8%) than men (64.8%; *p* = 0.041). No significant gender differences were found in marital status, ethnic group, or financial security. In terms of BMI, men were significantly more likely to be overweight, whereas women were more likely to be within the normal weight range (*p* < 0.001). The prevalence of PA in nature (at least once a week) during different seasons was highest in summer (n = 757; 81.9%), following spring (n = 712; 77.1%) and autumn (n = 698; 75.5%) and lowest in winter (n = 587; 63.5%). For total PA, similar tendencies in seasonality were observed. The frequency of respondents engaged in MVPA for at least 30 min once a week and more often was higher in the summer than in the winter season (n = 767; 83.0% and n = 672; 72.7%, respectively). Also, the total score of the LTEQ was significantly higher in the summer season as compared to the winter, accordingly (95.97 ± 54.74 vs. 77.57 ± 53.00, *p* < 0.001).

Seasonal patterns of PA in nature did not significantly differ by gender, education level, marital status, or place of residence (all *p* > 0.05, Table 2). However, financial security showed a strong association with PA patterns (χ^2^ = 22.33, df = 4, *p* < 0.001), with regular engagement in nature-based PA being more prevalent among those reporting higher financial security. For instance, 72.4% of participants in the “more secure” group engaged in regular PA in nature, compared to only 48.1% of the “less secure” group.

Next, a comparison of study measures in patterns of PA in nature seasonality is provided in Table 3. Specifically, participants in the group of more regular PA in nature demonstrated significantly greater nature exposure, green-space proximity, nature connectedness, and WHOQOL-BREF quality of life scores compared to those with irregular or seasonally variable activity (all *p* < 0.001). Age also differed significantly between groups (*p* = 0.011), with the regular PA group being older on average. No significant group differences were found for BMI or perceived restoration in nature.

The regression model significantly predicted overall quality of life, accounting for 30% of the variance (R^2^ = 0.30, Table 4). Higher financial security, greater nature exposure, perceived restoration, nature proximity, and nature connectedness emerged as significant positive predictors (all *p* < 0.05). Conversely, higher BMI, older age, and rural residence were associated with lower quality-of-life scores. Gender, marital status, education level, and regularity of PA in nature did not significantly predict overall quality of life when other factors were controlled.

## 4. Discussion

Considering the proven benefits of nature exposure on human health and the necessity of ongoing PA for these health effects, it is crucial to pinpoint factors that influence seasonal changes in PA within natural settings. The study aimed to evaluate how PA in natural environments varies with the seasons and explored its connections to demographic factors, nature-related variables, and quality of life among adult women and men. We hypothesized that PA in natural settings would be at its lowest during winter and peak in summer. Additionally, we anticipated that greater seasonal fluctuations in PA would be associated with lower socioeconomic status, proximity to green spaces, nature exposure, nature connectedness, perceived restorative effects of nature, and overall quality of life.

Generally, the findings of the present study confirmed both hypotheses. In our research, we observed the lowest rates of PA in nature in winter and the highest in summer. These findings are in accordance with previous studies reporting the same trends of PA seasonality ([11]; [15]; [29]; [63]; [64]; [66]). However, our study provides new empirical data that the seasonal variability of PA in natural environments (i.e., working, walking, leisure cycling, etc.) follows the same trend as general PA. A similar PA seasonality trend was observed in a study assessing PA behaviour in urban green spaces ([50]). An analysis of general PA trends showed that outdoor PA during colder and darker periods is also significantly lower compared to warm periods, and it suggests that PA in natural environments is possibly not substituted by indoor PA. In other words, it seems that the respondents of our sample do not have routines to adapt their PA habits to colder periods. This may be attributed to biological factors and poor weather conditions ([63]), but also to a lack of knowledge about winter PAs, the availability of comfortable winter outdoor apparel, a lack of winter sports equipment, a reduced need for physical work around the home (e.g., gardening), and other factors that future studies are recommended to analyze in exploring why people are more sedentary in winter time. In Lithuania, people traditionally enjoy winter recreational activities such as skiing and skating. However, winters have become warmer and snow less frequent, potentially leading to a lack of tradition and knowledge regarding wintertime PA that is a part of Scandinavian countries living in a similar climate ([17]).

The second, our assumption was also confirmed. We found that lower financial security was associated with less regular and more seasonal PA in natural environments. This finding is in accordance with the results of a previous study, which reported that lower socioeconomic status and financial constraints are associated with greater seasonal variability of exercise ([13]; [25]). These results might be explained by evidence that while nature availability in living areas is associated with greater PA in natural environments ([6]; [45]), socially deprived groups usually live in places that are distant from natural environments and are perceived as less safe ([51]) and more polluted ([14]). Therefore, the PA in natural environments might be lower in individuals with lower financial security. However, our findings confront previous evidence suggesting that socio-economic status is not associated with the frequency of exercising in natural environments ([7]) and a higher seasonal variability of PA ([52]). These discrepancies may be due to differences in study methodology. Exercising in natural environments is free of charge, and therefore, it might be more available for socially deprived groups compared to organized sports and gyms. However, the results of our study suggest that it might be true for warmer seasons, but not for dark, wet, and cold periods. Future studies should further analyze this question.

In the present study, we found that those in the group involving the seasonal variability of PA in nature were younger, reporting a lower proximity of nature to their living residence, lower nature exposure and connectedness to nature, and a lower quality of life compared to the group of regular PA in nature. These results are in accordance with previous findings that report that green exercise is more prevalent in older adults ([7]), and the young generation reports lower nature connectedness ([3]). Previous studies also showed that nature proximity in the living area is related to greater PA ([45]). Also, these findings are partially in line with previous findings reporting that greater nature exposure is associated with higher PA ([22]; [53]). Our study adds new knowledge that nature proximity and nature exposure are associated with the lower seasonal variability of PA in nature. However, we observed no significant differences in perceived nature restoration between regular PA in natural environments and seasonal groups. This finding is difficult to explain due to the lack of similar studies. However, one explanation is that poor weather conditions may reduce potential recovery and outweigh the motivation to be physically active in natural environments. However, these assumptions are speculative, and future studies are recommended to further investigate these associations.

Additionally, in the present study, we also explored how the seasonality of PA in nature and other analyzed variables were associated with quality of life. The analysis of the results showed that the main contributor to quality of life is financial security, followed by nature exposure and perceived restoration in nature, nature proximity, living in urban areas, and being younger. Less variability of PA in natural environments during seasons increased quality of life insignificantly. While this was not a primary aim of the study, these results add important insights. Specifically, our findings show that higher socioeconomic status and possibilities to experience nature in the living environment significantly contribute to quality of life. Generally, our results are in accordance with previous studies suggesting that nature exposure is associated with a higher quality of life and well-being ([19]; [39]; [69]).

Taken together, our findings point to the problem of inequitable access to PA in natural environments during the darker, colder, and wetter seasons. It seems that inequitable access to PA in nature manifests as a higher seasonality of PA in natural environments. This is an important new finding, and it is recommended that future studies assessing the seasonality of PA in natural environments consider socioeconomic status. The study’s practical implications underscore the significance of addressing inequitable access to natural environments in promoting consistent PA during various seasons and enhancing overall well-being. Specifically, there is a need for targeted interventions to improve access to natural environments for individuals with lower financial security. One potential solution is the implementation of subsidized programmes or initiatives, which would facilitate greater access to PA in nature for economically disadvantaged groups. Previous studies concluded that access to green or natural open space is associated with better health of socially deprived populations ([37]). Programmes that focus on increasing nature exposure and PA among younger adults, particularly those with lower financial security, could help mitigate the negative impacts on their quality of life. Community-based solutions, such as regularly organized local nature clubs or outdoor activity groups, can help bridge the gap in nature exposure and PA for those living in urban areas with limited access to green spaces ([43]). Future studies should continue to explore the relationship between socioeconomic status and the seasonal variability of PA in natural environments, considering additional individual and environmental factors to develop more comprehensive strategies.

The present study is subject to several limitations that should be discussed. The primary constraint of this study is its cross-sectional design, which hinders the interpretation of the direction of the associations. Secondly, while the present study explored the seasonality of PA in nature, it did not specifically address the prevalence of indoor PAs across different seasons. Addressing these limitations in future studies is therefore recommended. Thirdly, the present sample exhibited a highly unequal distribution of men and women, with men constituting a mere 26.4% of the sample. Finally, the nature of the data collection method, namely, that it was self-reported rather than objectively measured, might have led to an overestimation or underestimation of PA levels, an issue that has been previously identified ([44]). Also, we did not evaluate the duration and social context of PA in natural environments; however, these factors might influence the results ([71]).

## 5. Conclusions

Given the established positive impact of nature exposure on human health and the importance of maintaining regular PA to achieve these benefits, it is essential to identify the factors that affect seasonal variations in PA within natural environments. The findings of the present study showed that the lowest rates of PA in natural environments are in winter and the highest in summer. In the present sample, the seasonal variability of PA in natural environments was associated with lower financial security, younger age, lower proximity of nature to the living residence, lower nature exposure and connectedness, and lower quality of life. These findings point to the problem of inequity and suggest that inequitable access to PA in nature manifests as a higher seasonality of PA in natural environments. The practical implications of the study highlight the importance of addressing inequitable access to natural environments to promote consistent PA throughout seasons and enhance overall well-being. During the dark, wet, and cold periods, it is important to implement targeted interventions that improve access to natural environments for individuals with lower financial security. This could help reduce inequity in physical activity.

## Figures and Tables

**Table 1 behavsci-15-00773-t001:** Comparison of sociodemographic and anthropometric characteristics by gender (n = 924).

Characteristics		Men (n = 244)		Women (n = 680)		*p*
		n	%	n	%	
Age, years (m ± SD)		41.8	14.0	39.4	11.7	0.016
Education	secondary	49	21.6	56	8.5	<0.001
	in full-time studies	28	12.3	43	6.6	
	undergraduate	95	41.9	290	44.2	
	postgraduate	55	24.2	75.3	40.7	
Marital status	single	54	22.8	116	18.5	0.269
	in committed relationship	46	19.4	113	18.1	
	married/in partnership	137	57.8	397	63.4	
Place of residence	urban	158	64.8	488	71.8	0.041
	rural	86	35.2	192	28.2	
Ethnic group	majority	216	88.5	626	92.1	0.124
	minority	7	2.9	20	2.9	
	not sure	21	8.6	34	5.0	
Financial security	less secure	42	17.2	116	17.1	0.205
	average, like most	149	61.1	450	66.2	
	more secure	53	21.7	114	16.8	
Body mass index	underweight (<18.5 kg/m^2^)	5	2.1	30	4.5	<0.001
	normal weight (18.5–24.9 kg/m^2^)	99	40.7	415	61.1	
	overweight (25.0–29.9 kg/m^2^)	102	42.0	159	23.4	
	obesity (≥30.0 kg/m^2^)	37	15.2	75	11.0	

**Table 2 behavsci-15-00773-t002:** Comparison of sociodemographic characteristics by seasonal patterns of physical activity in nature (n = 924).

Groups	Irregular or no PA in Nature, n = 153		Seasonal Variability, n = 201		Regular PA in Nature, n = 570		Total		χ^2^, df, *p*
	n	%	n	%	n	%	n	%	
Gender									
Men	35	14.3	48	19.7	161	66.0	244	100.0	2.63, 2, 0.268
Women	48	17.4	153	22.5	409	60.1	680	100.0	
Education									
Secondary	18	17.1	24	22.9	63	60.0	105	100.0	1.51, 6, 0.959
In full-time studies	11	15.5	19	26.8	41	57.7	71	100.0	
Undergraduate	61	15.8	82	21.3	242	62.9	385	100.0	
Postgraduate	55	17.1	68	21.1	199	61.8	322	100.0	
Marital status									
Single	33	23.1	21	14.7	89	62.2	143	100.0	5.70, 4, 0.223
In a committed relationship	32	17.7	43	23.8	106	58.5	181	100.0	
Married/in partnership	170	19.5	95	17.6	339	62.9	539	100.0	
Place of residence									
Urban	111	17.2	151	23.4	384	59.4	646	100.0	4.86, 2, 0.088
Rural	42	15.1	50	18.0	186	66.9	278	100.0	
Financial security									
Less secure	39	24.7	43	27.2	76	48.1	158	100.0	22.33, 4, <0.001
Average, like most	92	15.4	134	22.4	373	62.3	599	100.0	
More secure	22	13.2	24	14.4	121	72.4	167	100.0	

Note. PA—physical activity.

**Table 3 behavsci-15-00773-t003:** A comparison of the study measures (m ± SD) by the seasonal patterns of physical activity in nature (n = 924).

Variables	Irregular or no PA in Nature (n = 153)	Seasonal Variability (n = 201)	Regular PA in Nature (n = 570)	*p*
Age, years	38.99 ± 12.18	38.11 ± 12.21	40.95 ± 12.44 ^b^	0.011
Body mass index, kg/m^2^	25.02 ± 5.15	24.39 ± 4.23	24.82 ± 4.61	0.398
Physical activity in nature	1.27 ± 0.35	2.95 ± 0.69 ^a^	4.31 ± 0.77 ^ab^	<0.001
Green space proximity	3.27 ± 1.21	3.46 ± 1.25	3.74 ± 1.12 ^ab^	<0.001
Nature exposure	3.69 ± 0.87	3.93 ± 0.72 ^a^	4.26 ± 0.63 ^ab^	<0.001
Nature connectedness	3.60 ± 0.79	3.78 ± 0.63	3.88 ± 0.71 ^a^	<0.001
Perceived restoration in nature	5.22 ± 1.60	5.32 ± 1.31	5.47 ± 1.53	0.141
Quality of life	64.59 ± 14.68	67.62 ± 13.23	70.82 ± 13.47 ^ab^	<0.001

Note. PA—physical activity. a: *p* < 0.05 compared to the irregular or no physical activity in nature group; b: *p* < 0.05 compared to the seasonal variability group.

**Table 4 behavsci-15-00773-t004:** Multiple regression analysis predicting overall quality of life (n = 924).

Characteristics	B	SE	β	t	*p*
Women (ref. men)	−0.36	0.96		−0.37	0.710
Age, years	−0.09	0.04	−0.08	−2.27	0.024
Body mass index, kg/m^2^	−0.43	0.10	−0.14	−4.46	<0.001
Rural residence (ref. urban)	−2.82	0.94		−3.01	0.003
Education: full-time studies (ref. secondary)	0.94	1.97		0.48	0.635
Education: undergraduate (ref. secondary)	1.36	1.40		0.97	0.332
Education: postgraduate (ref. secondary)	1.02	1.47		0.70	0.487
Marital status: in a committed relationship (ref. single)	2.11	1.32		1.60	0.110
Marital status: married/cohabiting partnership (ref. single)	1.41	1.09		1.30	0.194
Financial security: like most (ref. not secure)	8.15	1.14		7.16	<0.001
Financial security: more secure (ref. not secure)	13.68	1.44		9.53	<0.001
Nature proximity	1.22	0.36	0.10	3.37	<0.001
Nature exposure	3.24	0.68	0.17	4.75	<0.001
Nature connectedness	1.46	0.74	0.08	1.98	0.048
Perceived restoration in nature	1.61	0.34	0.17	4.75	<0.001
Regular PA in nature: all seasons (ref. non or irregular)	1.19	0.89		1.34	0.181
Model summary: R = 0.55; R^2^ = 0.30					

Note. ref.—reference; PA—physical activity; B—unstandardized regression coefficient; SE—standard error; β—standardized regression coefficient calculated only for continuous variables; t—*t*−test.

## Data Availability

The original contributions presented in this study are included in the article/Appendix A. Further inquiries can be directed to the corresponding author.

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
