# Peer review of "An Analysis of the Factors Associated with the Seasonal Variability of Physical Activity in Natural Environments in a Sample of Lithuanian Adults"

_behavsci, 2025, doi:10.3390/bs15060773_

Round 1
Reviewer 1 Report
Comments and Suggestions for Authors
Dear Authors,
thank you very much for the interesting read. I reccomend it for publication after some minor revisions.
Methodology: l. 189 following. A bit more explanation is needed: The invitation to participate were related to the municipal Facebook Pages? Explain a bit more in detail - did you also consider different sizes or geographic locations to reach out to achieve a broader range of participants?
Looking at the resulta and what you got, it would be interesting to refelect a bit about what response you collected. For example, the survey outcome shows that more women participated. Some words reflecting the sample of respondents would be useful in the Results section.
l.162 – Conclusions:
In the introduction section, you state that your data might help to inform policies related to decreasing health inequalities. So, for the conclusion section, it would be nice to pick up this aspect in a bit more depth again. Here, the only sentence so far is: “Practical implications of the study highlight the importance of addressing inequitable access to natural environments to promote consistent PA throughout seasons and enhance overall well-being” – perhaps you could add some more reflections here so it nicely links with the introduction and give some brief reccomendations here.
Author Response
Review 1
Dear Authors,
thank you very much for the interesting read. I reccomend it for publication after some minor revisions.
Thank you for reviewing our article and for your positive feedback.
Methodology: l. 189 following. A bit more explanation is needed: The invitation to participate were related to the municipal Facebook Pages? Explain a bit more in detail - did you also consider different sizes or geographic locations to reach out to achieve a broader range of participants?
Thank you, we revised the text as follows: “The survey link was distributed via a sponsored Facebook advertisement targeting all major municipalities across the country and addressing their sizes”.
Looking at the resulta and what you got, it would be interesting to refelect a bit about what response you collected. For example, the survey outcome shows that more women participated. Some words reflecting the sample of respondents would be useful in the Results section.
We provided a specific sample representing information in the first paragraph of the Results section.
l.162 – Conclusions:
In the introduction section, you state that your data might help to inform policies related to decreasing health inequalities. So, for the conclusion section, it would be nice to pick up this aspect in a bit more depth again. Here, the only sentence so far is: “Practical implications of the study highlight the importance of addressing inequitable access to natural environments to promote consistent PA throughout seasons and enhance overall well-being” – perhaps you could add some more reflections here so it nicely links with the introduction and give some brief reccomendations here.
Thank you, we revised conclusions adding this sentence: “During the dark, wet, and cold periods, it is important to implement targeted interventions that improve access to natural environments for individuals with lower financial security. This could help reduce inequity in physical activity.”
Reviewer 2 Report
Comments and Suggestions for Authors
The topic is extremely relevant and current, especially in the context of public health and discussions about urban green spaces and well-being. The originality lies in the proposal to relate quality of life data with the regular practice of physical activity (PA) in natural environments throughout the seasons, something little explored. The gap addressed is clear: most previous studies focused on PA in general, without differentiating between natural and artificial environments, nor taking into account the influence of climatic and social variables.
This study offers several important contributions, as it empirically demonstrates the relationship between seasonal variability of PA and social inequality, suggesting that factors such as financial security and access to nature modulate physical behavior throughout the year. It uses a robust set of instruments (WHOQOL-BREF, CNS, NES, ROS, LTEQ), with excellent psychometric reliability. It suggests that health promotion programs should consider the seasonal component, especially in temperate countries, which has an important impact on the planning of feasible intervention strategies with and in nature.
Although the study is well structured, there are points that deserve attention, and that are not necessarily an impediment to publication, but that could be emphasized as limitations of the study:
- The design is cross-sectional, which limits causal inferences. A longitudinal follow-up would be more suitable to capture real changes throughout the year.
- Data collection was done by online self-reporting and non-probabilistic sampling, which can introduce selection bias and social desirability.
- The sample has a gender imbalance (73% women), which may limit the generalization of the findings.
- Although it evaluates the frequency of PA in the seasons, it does not adequately distinguish between outdoor and indoor activities during the winter, which would be relevant.
The conclusions are consistent. References are abundant, current, and adequate, with extensive citation of systematic reviews, cross-national studies, and WHO guidelines. The bibliography demonstrates a solid theoretical basis, especially on the psychosocial benefits of contact with nature and restoration models (ART and SRT).
This is a rigorous, well-structured study with important implications for public policies and urban health. By connecting social inequalities to the practice of outdoor physical activity, especially in winter, the article contributes significantly to the debate on equitable access to nature and the promotion of healthy lifestyles.
Author Response
Review 2
The topic is extremely relevant and current, especially in the context of public health and discussions about urban green spaces and well-being. The originality lies in the proposal to relate quality of life data with the regular practice of physical activity (PA) in natural environments throughout the seasons, something little explored. The gap addressed is clear: most previous studies focused on PA in general, without differentiating between natural and artificial environments, nor taking into account the influence of climatic and social variables.
Thank you for reviewing our article and for your positive feedback.
This study offers several important contributions, as it empirically demonstrates the relationship between seasonal variability of PA and social inequality, suggesting that factors such as financial security and access to nature modulate physical behavior throughout the year. It uses a robust set of instruments (WHOQOL-BREF, CNS, NES, ROS, LTEQ), with excellent psychometric reliability. It suggests that health promotion programs should consider the seasonal component, especially in temperate countries, which has an important impact on the planning of feasible intervention strategies with and in nature.
Thank you.
Although the study is well structured, there are points that deserve attention, and that are not necessarily an impediment to publication, but that could be emphasized as limitations of the study:
- The design is cross-sectional, which limits causal inferences. A longitudinal follow-up would be more suitable to capture real changes throughout the year.
- Data collection was done by online self-reporting and non-probabilistic sampling, which can introduce selection bias and social desirability.
- The sample has a gender imbalance (73% women), which may limit the generalization of the findings.
- Although it evaluates the frequency of PA in the seasons, it does not adequately distinguish between outdoor and indoor activities during the winter, which would be relevant.
Thank you. All these limitations of the study are included in the last paragraph of the Discussion section.
The conclusions are consistent. References are abundant, current, and adequate, with extensive citation of systematic reviews, cross-national studies, and WHO guidelines. The bibliography demonstrates a solid theoretical basis, especially on the psychosocial benefits of contact with nature and restoration models (ART and SRT).
This is a rigorous, well-structured study with important implications for public policies and urban health. By connecting social inequalities to the practice of outdoor physical activity, especially in winter, the article contributes significantly to the debate on equitable access to nature and the promotion of healthy lifestyles.
Thank you.